# Semi-Supervised Learning to Automate Tumor Bud Detection in Cytokeratin-Stained Whole-Slide Images of Colorectal Cancer

**DOI:** 10.3390/cancers15072079

**Published:** 2023-03-30

**Authors:** John-Melle Bokhorst, Iris D. Nagtegaal, Inti Zlobec, Heather Dawson, Kieran Sheahan, Femke Simmer, Richard Kirsch, Michael Vieth, Alessandro Lugli, Jeroen van der Laak, Francesco Ciompi

**Affiliations:** 1Department of Pathology, Radboud University Medical Center, 6525 GA Nijmegen, The Netherlands; 2Institute of Tissue Medicine and Pathology, University of Bern, 3008 Bern, Switzerland; 3UCD School of Medicine, St. Vincent’s University Hospital, D04 T6F4 Dublin, Ireland; 4Division of Pathology and Lab Medicine, University of Toronto, Toronto, ON M5S 1X5, Canada; 5Klinikum Bayreuth, Friedrich-Alexander-University Erlangen-Nuremberg, 91054 Bayreuth, Germany; 6Center for Medical Image Science and Visualization, Linköping University, 581 83 Linköping, Sweden

**Keywords:** deep learning, computational pathology, colorectal carcinoma, tumor budding, object detection

## Abstract

**Simple Summary:**

Tumor budding is a promising and cost-effective histological biomarker with strong prognostic value in colorectal cancer. It is defined by the presence of single tumor cells or small clusters of cells within the tumor or at the tumor-invasion front. Deep learning based tumor bud assessment can potentially improve diagnostic reproducibility and efficiency. This study aimed to develop a deep learning algorithm to detect tumor buds in cytokeratin-stained images automatically. We used a semi-supervised learning technique to overcome the limitations of a small dataset. Validation of our model showed a sensitivity of 91% and a fairly strong correlation between a human annotator and our deep learning method. We demonstrate that the automated tumor bud count achieves a prognostic value similar to visual estimation. We also investigate new metrics for quantifying buds, such as density and dispersion, and report on their predictive value.

**Abstract:**

Tumor budding is a histopathological biomarker associated with metastases and adverse survival outcomes in colorectal carcinoma (CRC) patients. It is characterized by the presence of single tumor cells or small clusters of cells within the tumor or at the tumor-invasion front. In order to obtain a tumor budding score for a patient, the region with the highest tumor bud density must first be visually identified by a pathologist, after which buds will be counted in the chosen hotspot field. The automation of this process will expectedly increase efficiency and reproducibility. Here, we present a deep learning convolutional neural network model that automates the above procedure. For model training, we used a semi-supervised learning method, to maximize the detection performance despite the limited amount of labeled training data. The model was tested on an independent dataset in which human- and machine-selected hotspots were mapped in relation to each other and manual and machine detected tumor bud numbers in the manually selected fields were compared. We report the results of the proposed method in comparison with visual assessment by pathologists. We show that the automated tumor bud count achieves a prognostic value comparable with visual estimation, while based on an objective and reproducible quantification. We also explore novel metrics to quantify buds such as density and dispersion and report their prognostic value. We have made the model available for research use on the grand-challenge platform.

## 1. Introduction

Tumor budding is characterized by the presence of isolated single cancer cells or clusters of up to four cancer cells of epithelial origin. This phenomenon is widely recognized as a prognostic biomarker, predicting lymph node metastasis, disease progression, and unfavorable survival in colorectal cancer (CRC), among others [1,2]. In 2016, a standardized reporting method of assessing tumor budding in CRC was established (ITBCC, 2016) [3]. According to this recommendation, pathologists should scan the invasive tumor front at low magnification and select the area with the highest tumor bud density, a circular ‘hotspot’ with a radius of 0.5 mm.

Next, they should identify and count the tumor buds at higher magnification and convert the count into a score according to a three-tier system (Bd1 -low budding- 0–4 buds; Bd2 -intermediate budding- 5–9 buds; Bd3 -high budding- 10 or more buds). Traditionally, this procedure is performed on routinely available H&E-stained tissue sections. However, peritumoral inflammatory infiltration may mask tumor buds, making it challenging to detect them on H&E. Additionally, tumor buds and reactive stromal cells can occasionally be difficult to distinguish in H&E.

Under these circumstances, an immunohistochemical (IHC) pan-cytokeratin (CK) stain can be used to highlight the epithelium versus activated fibroblasts and immune cells (see Figure 1). Therefore, IHC is frequently used to help pathologists identify tumor buds (TBs) [2]. Several studies involving experienced gastrointestinal pathologists have reported the prognostic value of TB based on H&E staining, as well as on IHC [3,4,5]. At the same time, moderate-to-substantial inter-observer variability has been reported for TB scoring in both H&E and IHC, including our recent work [6]. This variability can potentially increase when non-subspecialist gastrointestinal pathologists are involved, especially when H&E slides are inspected.

Concerns over interobserver variability as well as the labor-intensive aspect of manual TB detection suggest the potential role of a computer-aided approach to TB quantification. The advent of digital or computational pathology allows the development of computer algorithms that can improve the reproducibility of the tumor bud identification process. In particular, convolutional neural network (CNN) systems have shown their ability to match or even exceed the performance of human experts in medical object recognition and classification tasks. Several authors studied the development of an automatic or semi-automatic tumor bud detection method, reviewed in a post-ITBCC review in 2021 [7]. IHC can help assess tumor buds more easily by the effect of the CK stain, which stains the epithelium and removes other single-cell objects, e.g., activated fibroblasts. For this reason, attempts have predominantly been made in CK-stained tissue.

### 1.1. Related Work

Several computer methods have been proposed to address the detection of tumor buds in IHC. Most methods usually require a manual selection of a region of interest (ROI), typically selected at specific predetermined locations, e.g., the invasive front region, or by explicitly excluding areas such as necrosis. Within these ROIs, tumor bud detection is achieved using classical image analysis operations, sometimes in combination with some form of machine learning. Fischer et al. [4] used QuPath software to classify tumor buds in digitized CRC tissue microarrays (TMAs) based on the color and measured cluster size (area instead of nuclei number) and compared the results with manually obtained tumor bud yields in H&E and CK per count and in a bud-by-bud fashion. They analyzed the impact of all methods on patient survival. They found a strong correlation between manual CK and semi-automated methods and, for each method, reduced survival at higher tumor bud scores. However, TB selection based on size might result in false-positive detections for compact tumor clusters of up to four cells. To overcome this problem, Fauzi et al. [8] followed a similar approach but added nuclei detection to reduce the number of false positives. Methods that rely on stain intensity might not generalize well to the stain variations found in clinical practice. These methods use predetermined constants that are often fine-tuned on a dataset with limited variation. CNNs can be trained using stainaugmentations [9] to overcome this. Therefore, Bergler et al. [10] proposed a two-step hybrid approach, in which possible TB candidates are first selected by color and minimum size using image processing methods as described above, after which a CNN is used to achieve a reduction in false-positive candidates. Weis et al. [11] followed a similar path, first using image processing steps for the pre-selection of TB candidates on a CK-positive area and size, after which CNN training takes place and validation is performed on 20 TMA cores, manually labeled by a pathologist. The network trained in this way is used to post-classify the real TB against any lookalikes. In the test phase performed on whole slide images, this author finds a connection between the number of significant hotspot locations and the latter parameter in a small group of patients. However, no significant correlations between the TB number(-derivatives) or TB score and nodal status.

### 1.2. Our Contribution

Although tumor budding scoring by pathologists is primarily performed in H&E, IHC still plays a vital role in clinical routine as it helps pathologists identify TB in difficult (inflammatory) regions. Therefore, an IHC tumor bud detection tool is desirable, to (1) assist pathologists in clinical routine of the detection of TB; (2) as a research tool; and (3) as an initial step for the development of an algorithm for TB detection in H&E, as it will provide an indirect reference material for this purpose. In this work, we introduce several contributions to the field of automated TB detection in the IHC-stained whole-slide images of CRC patients. We address the problem of limited availability of manually annotated tumor buds to train computer algorithms, given that it is an expensive procedure subject to high inter-observer variability. We leveraged a collection of n = 1765 manually annotated objects which were either tumor bud or non-tumor bud from a previous study [6], representing a set of sparse annotations (i.e., not all objects of interest were annotated) at the WSI level. This scenario poses problems when training traditional object detection models, which typically need exhaustive manual annotations of objects to detect during training. To address this problem, in this work, we use a semi-supervised learning approach to train object detection models using sparsely annotated data, analogous to Niazi et al. [12] who utilized stain deconvolution to create a reference standard directly based on the positive cytokeratin objects. As cytokeratin highlights all epithelium and not just TBs, here we use a small sparsely annotated dataset of TBs and non-TBs. The proposed approach is based on training two deep learning networks and it makes it possible to combine the advantages of a small, labeled dataset to guide the learning process and a large, exhaustive, pseudolabeled dataset to increase the robustness to variations in object appearance, without the need for time-consuming exhaustive manual annotations. Because the second model uses the information learned from the first, we will refer to this method as a teacher–student approach. Although the teacher–student is commonly associated with knowledge distillation, where the teacher and students are not identical in the complexity of the model, in this work, we propose to keep this terminology to emphasize the sequential nature of learning and knowledge transferred from the first model to the second model. As a result, the proposed method allows training computer models to detect TBs in any region of the slide, de facto stepping away from TB detection in pre-selected regions and allows a complete analysis at the whole-slide image level. Thanks to this characteristic, in this work, we also propose a method to compute a tumor bud density map at the WSI level, which allows to (1) identify the TB hotspot and extract TB count as a biomarker, in line with the ITBCC guidelines; (2) explore novel biomarkers based on the spatial heterogeneity of TB distributions, and investigate their predictive value, which represents additional contributions of our work. We validated every step of the proposed approach (TB detection, hotspot detection, prognosis) using manual annotations from a panel of experienced pathologists and clinical and survival data from a cohort of n = 40 patients.

Finally, we make the algorithm publicly available for research purposes as a stand-alone application running on the grand-challenge.org platform (https://grand-challenge.org/algorithms/colon-budding-in-ihc/, accessed on 1 February 2023). As such, we envision that this algorithm will serve as a testing tool for researchers and clinicians in the computational pathology field, which can be used for exploratory purposes and for comparison with other algorithms.

## 2. Materials and Methods

### 2.1. Materials

Two sets of resected CRCs were collected. The first set was used to develop and validate the tumor bud detection model(s). The second set was collected to test the deep learning model on (cor-)relations with manual TB counts and hand-selected hotpots for patient survival analyses. In the next section, both datasets are described in detail. An overview of the datasets can be found in Table 1.

#### 2.1.1. Model Development Data

In a previously published study on human visual budding scoring, we used 84 tissue sections taken from 45 patients [6]. Six of these slides were IHC-stained with AE1/AE3 and scanned at the Institute of Pathology, University of Bern (Switzerland), two were stained with AE1/AE3 immunohistochemistry and scanned at the Dublin University Hospital (Ireland). The remaining 76 slides were stained with CK8-18 immunohistochemistry and scanned at the Radboud University Medical Center, Nijmegen (The Netherlands).

In our previous work, all slides were scanned with a Pannoramic P250 Flash II scanner (3D-Histech, Hungary) using a 40× objective lens (yielding a specimen-level pixel-size of 0.24 × 0.24 μm). Digital image analysis was used to pre-select 3000 bud candidates, which were subsequently independently classified by seven experienced pathologists from the ITBCC as either (1) TB; (2) poorly differentiated cluster (PDC, defined as clusters of five or more tumor cells without gland formation); or (3) neither, in accordance with current definitions. For the purposes of this study, we grouped PDCs and objects labeled as neither group.

After calculating the majority votes, this resulted in a reference set of 1010 buds and 755 non-bud objects. In the current study, we used a subset of n = 74 whole slide images for training and model optimization. The remaining WSIs (n = 10)—one from Dublin, two from Bern, and eight from Nijmegen—were used as a multi-centric test-set. This test set contains 330 TB and 283 non-TB candidates. When composing these subsets, we ensured that all WSIs from a single patient were included in the same subset. In the following, we will use these codes for the three aforementioned train, validation, and test subsets, respectively: dataset dev-l(earning), dataset dev-v(alidation), dataset dev-t(est).

#### 2.1.2. Validation Data

From a cohort of 40 CRC patients from the Institute of Pathology, the University of Bern n = 240 tissue sections and associated clinical pathology reports were obtained. At Bern University Labs, the sections were stained with AE1/AE3 and CD8 to highlight the epithelium and cytotoxic t-cells, respectively. DAB staining was used for the cytokeratin stains and a red dye was used for CD8; however, in practice this red dye does not hamper the results of the algorithm as it was trained on DAB. Afterwards, the slides were scanned with a Pannoramic P250 Flash II scanner (3D-Histech, Hungary) at a spatial resolution of 0.24 μm/px. In all WSIs, a tumor budding hotspot and tumor bud count were established by an expert according to the ITBCC recommendations. This dataset is referred to as dataset eval.

### 2.2. Model Development

We hypothesized that the performance of a detection network would be adversely affected by incomplete (i.e., sparse) annotations, something that classification networks do not suffer from. To see the extent to which the expected adverse effect, we included a detection network alongside a classification network. The two deep learning networks were trained solely on our small, expert-labeled dataset dev-l, and were used to identify the additional candidates for training the student network. We will hereafter refer to these networks as the teacher networks.

The best performing teacher network was applied to the WSIs of the dataset dev-l and dev-v, after which the candidates detected by the teacher model were added to the dataset dev-l and dev-v. Thereafter, two new deep learning networks, called student, will be trained on the combined dataset. A schematic overview of the method can be found in Figure 2.

#### 2.2.1. Sparsely Annotated Data and Object Detection

Classification networks are able to predict whether an image contains a specific object or not, but provide no further information as to the location of the classified object. Detection networks, on the other hand, provide information about the class and location. Therefore, a detection network, such as faster R-CNN [13], YOLO [14], and SSD [15], is usually the first and best choice for a detection task, such as tumor bud detection. We selected the faster R-CNN network because it is known to perform well in the medical image analysis field [16,17,18,19]. As with most region-proposal-based network architectures, faster R-CNN works in a two-step fashion. Regions of interest are proposed in the region proposal part of the network at first, and then, in a second step, objects in these regions of interest are classified. During training, the locations of the proposed regions and the object classification within these regions are fine-tuned based on the reference standard. Because of this architecture, these network types are less suitable for sparsely annotated inputs, as objects without labels are falsely assumed to be negatives/background. With sparsely annotated data, this leads to a significant number of potential positive objects (unannotated TB) in data for training that are marked as negatives, which may deteriorate the network performance. In connection with the above, we also opted for a fully convolutional DenseNet network as an alternative teacher network architecture. The fully convolutional network is trained as a patch classifier, solely sampling patches from annotated regions, and labeling those patches with a single value based on the annotation.

#### 2.2.2. Student Development

The teacher network with the highest sensitivity on dataset dev-v was used to identify the additional candidates for training the student networks. This network was applied to the entire WSIs, without excluding specific regions. To maximize the precision of our model, only candidates for which the model gave a likelihood of >0.80 were included in the combined dataset. To further reduce the amount of mislabeled objects outside the DAB positive pixels, we used color deconvolution [20] to isolate the DAB-positive image pixels, which were grouped to form binary objects. This binary mask was used to remove pseudo-labeled buds when not present in the DAB mask. The new training dataset, dev-l+, now containing not only the initial n = 1200 candidates but also approximately 20,000 pseudo-labeled candidates. Because we did not pre-select any regions, no bias was introduced in the network by the manually selected objects from the previous study. Therefore, this network should be able to cope with all the cell types found in the entire WSIs, including the cell types found outside the tumor region. This ensures that the network is applicable at the WSI level without the need to manually pre-select a region. Due to the expected greater number of false positives among the pseudo-labeled tumor buds, these candidates were given a lower certainty (0.80) of representing a bud (the certainty of 0.8 was empirically found). In contrast, manually selected buds received a certainty of 1.0, which means that, during the training, a student network was less penalized when making a mistake with a pseudo-labeled object compared to the manually labeled data.

We trained a faster R-CNN and a DenseNet as student networks. These networks were trained on dataset dev-l+ and validated during training on dataset dev-v.

#### 2.2.3. Training Parameters

The input for the faster R-CNN network and DenseNet network was an RGB patch of, respectively, 1024 × 1024 px and 512 × 512 px (pixel size 1 μm). To enable the classification of arbitrary input sizes at test time, the average-pool layer of the DenseNet models were removed and the final convolutional layer was replaced with two 1 × 1 filters that represent each output class. During training, the random flipping, rotation, elastic deformation, blurring, brightness (random gamma), color, and contrast change augmentations were used for data augmentation. An adaptive learning rate scheme was used where the learning rate was initially set to 1 × 10^−4^ and then multiplied by a factor of 0.5 after every 25 epochs if no increase in performance was observed on the validation set. All networks were initialized with pre-trained ImageNet weights. The mini-batch size was set to one instance per batch, and the networks were trained for a maximum of 600 epochs, with 200 iterations per epoch. The training of the networks was stopped when no improvement of the validation loss was found for 50 epochs. The output of all networks with the DenseNet architecture is in the form of *C* likelihood maps. The arg-max of these likelihood maps was taken to obtain a final detection output. The output of the faster R-CNN network is all region proposals with a corresponding probability of it belonging to class *C*.

#### 2.2.4. Automated Hotspot Selection

In contrast to manual TB assessment, in which (according to ITBCC protocol) hotspot selection precedes tumor bud counting, in the automated procedure, all tumor buds are first detected and recorded in terms of location, after which the hotspots were selected. To identify the number of tumor buds within every potential hotspot, the following procedure was applied (Algorithm 1).
**Algorithm 1** Create tumor budding density map.**Ensure:** The network was applied to the entire slide1:**for** every pixel in the WSI **do**         ▹ identified by its x and y coordinate2:   Draw a circle with an area of 0.785 mm^2^.3:   Count the number of tumor buds within each circle.4:   Note the number of tumor buds within the circle on each (center) pixel of this circle.5:**end for**6:Normalize the density map based on the total number of tumor buds per slide.

This density map can also be visually displayed as a heat map of the whole slide to facilitate the visual identification of the region with the highest tumor buds.

Based on the density map, we could extract the regions with the highest amount of tumor buds within the hotspot. We simply cannot select (center-)pixels with the highest TB density, as the related hotspot areas might overlap. To prevent this, we used the following steps within the whole slide image: (1) the location with the highest tumor bud number within the whole slide image was selected; and (2) we label any pixel within the selected hotspot area of a TB with a value of 0. By doing this, the next hotspot has a minimum distance of the radius of the hotspot, resulting in a maximum overlap of 50% between the hotspots; (3) repeat steps 1 and 2 until 10 locations are found.

We determined the overlap between the manually and ten automatically selected hotspots per slide using the dice coefficient. If a dice score of 0.7 or higher was observed, we registered that as a good overlap between the two hotspots.

#### 2.2.5. Tumor Bud Distribution

The density map reflects how the tumor buds are distributed throughout the tissue. We hypothesize that tumor bud distribution might be a predictor for patient outcome. Tumor bud spread can be quantified in terms of entropy, which can be defined by a measure of system heterogeneity. Shannon’s entropy was calculated for every density map of the dataset eval.

#### 2.2.6. Statistical Analysis

Statistics were performed on dataset Val using R 4.1.3 (R Foundation College Station, TX, USA). The Cox proportional hazards regression models were performed in univariate and multivariate settings to calculate the hazard ratios (HRs). Multivariate adjustments were age (<65, ≥65), sex (male, female), T (1 and 2, 3 and 4), N (0, 1, and 2). Sixteen patients with insufficient follow-up data were removed from this analysis. A complete overview of the patient information can be found in Table 2.

Survival analysis was performed on three different budding counts, all on continuous values: (1) tumor bud counts by the expert; (2) tumor bud counts by the detection system in the same hotspot; and (3) tumor bud counts in the automatically selected hotspot by the algorithm.

We applied cut-off values to categorize the continuous tumor bud values for clinical decision making. For the tumor bud counts in the manually selected hotspots, we used the ITBCC (H&E) cut-off values. We applied the median as a cut-off value for the fully automated assessment. For Shannon’s entropy, we also applied the median value as a cut-off value.

## 3. Results

### 3.1. Detection Model Performance

The four different AI models were assessed on 613 tumor bud/non-tumor-bud candidates extracted from dataset dev-t. For these 613 candidates, we can only identify true positives and false negatives, resulting in a sensitivity score. Given the nature of the dataset and the training method, we have many detected objects without a reference label, which we cannot identify as true/false positives. Because of this, we cannot calculate a specificity.

The DenseNet model obtained the highest sensitivity score from the teacher models (0.83). The teacher faster R-CNN model scored worse, with a value of 0.47. All student models reached a higher sensitivity compared to the teachers’ networks. The student faster R-CNN model performed the best with a score of 0.91. The student DenseNet obtained a score 0.87. An overview of the different scores can be found in Table 3. The results of the different models are depicted in Figure 3.

### 3.2. Automatic vs. Manual Tumor Bud Count

The faster R-CNN student network was applied to the n = 240 slides of dataset eval, which had already been provided with the hotspot location and corresponding tumor bud count by an expert, conforming to the ITBCC protocol. We compared the manually obtained tumor bud counts with the automatically detected tumor bud numbers from the same hotspot area location using the Pearson correlation coefficient (Figure 4). The faster R-CNN network showed a correlation of 0.72 with the manually detected tumor buds. The network detected an average of 30% more tumor buds than the expert (Figure 5B).

### 3.3. Automatic vs. Manual Hotspot Detection

We found that the manually selected hotspot areas corresponded in 48% of the WSIs with the top three of the automatically chosen areas, in 72% of the cases with one of the top ten of these areas and in 28% of the cases with none of the automatically generated hotspots Figure 5C,D. Examples of density maps used for the hotspots can be seen in Figure 5A.

### 3.4. Survival Analysis

No relationship was observed between the budding (assessed either visually or by the automated method) and the overall survival (HR = 1.06; *p* = 0.06 and HR = 1.03; *p* = 0.31, respectively) in univariable Cox regression in the manually selected hotspots. A hazard-ratio of 1.56 (*p* = 0.55) was found with the fully automated method. Shannon’s entropy over the density maps resulted in a hazard ratio of 3.96 (*p* = 0.03). The multivariate analysis resulted in hazard ratio values of 1.07 (*p* = 0.05), 1.02 (*p* = 0.63), 1.63 (0.61), 3.96 (*p* = 0.06) for the manual count, automatic count in the same hotspot, fully automated method, and Shannon’s entropy, respectively.

## 4. Discussion

Tumor budding is a well-established prognostic feature in CRC in several clinical settings [2]. Despite substantial evidence, tumor bud assessment is still not part of routine clinical practice. One of the main contributing factors is intra-observer variability; agreement on the individual tumor buds in IHC is only moderate. To overcome this, we focused on developing an algorithm for TB detection in IHC-stained WSIs. The data [5] have shown the vital role of IHC in clinical practice to identify the tumor buds in difficult regions. Therefore, we envisioned the role of the proposed method as a support tool for both research and clinical practice, potentially as a complementary system to increase the efficiency and reproducibility of TB scoring.

One of the essential elements for developing a deep learning network is a large and high-quality labeled dataset, which is often difficult to obtain. To overcome this limitation, we propose leveraging a large set of pseudo-labeled data using a semi-supervised learning technique referred to as the teacher–student approach in this paper. Using this technique, we showed that, across the board, all student methods obtained a higher performance compared to their teacher counterparts (Table 3), which shows the potential of using pseudo-labeled data, which could improve the robustness. Additionally, this method allows us to train detection networks that typically need the exhaustive manual annotations of objects to detect during training. Because the quality of the pseudo-labeled data might be lower, a ‘human in the loop’ approach could be used to overcome this.

In line with the described semi-supervised method, two different networks (faster R-CNN and DenseNet) were trained as teacher models and subsequently the same types of networks were trained on the output of the best teacher. DenseNet was chosen as it is trained as a patch classifier that is unaffected by sparse annotations. Our results show that it indeed outperforms the faster R-CNN network as a teacher. The lower performance of DenseNet after training on the extended dataset may be related to its lower output resolution, an inherent limitation in this model that results in overlapping objects. Since we are using IHC-stained slides, this can be solved by multiplying the DenseNet output with the CK-positive channel of the image. However, this approach would move the problem to an accurate separation of signals (i.e., hematoxylin and DAB) in immunohistochemistry, potentially introducing additional noise (i.e., when the stain is weak) and is therefore suboptimal. As a detection network, we selected faster R-CNN because of the higher performance compared to other, faster popular detection networks, such as SSD [15]. However, for smaller objects, such as tumor buds, SSD performs significantly worse than faster R-CNN [21]. As speed was not our primary concern, we opted for faster R-CNN. This allows us to process the entire whole-slide image in approximately six minutes.

Typical detection networks, such as faster R-CNN, are less suited for training on sparse annotated data. The DenseNet performed better in the teacher training, but in the student training of the two networks on the full, pseudo-labeled dataset, the faster R-CNN detection network had the highest sensitivity of 0.91. Bergler et al. [10] achieved a similar sensitivity (0.93) with a hybrid method; in a first step, all cytokeratin-positive objects are selected, followed by a CNN in a second step to identify the true tumor buds. An inherent problem of this hybrid method is the hypersensitivity to stain variation. With our one-step approach, we can achieve greater robustness against these stain variations.

In the training set of the student networks, artifacts (such as cytokeratin remnants) are likely present as negative examples, which can improve the robustness of the model, and therefore make it capable of dealing with the multiple sources of noise that can be found in the images. In the current versions of all networks, but to a greater extent with the student models, there is an increase in false-positive classification at the large necrotic tissue parts, as we have not specifically trained in these areas. Other methods have specifically excluded these kinds of regions. With our approach, this could be addressed by adding examples with the manual annotations of necrotic tissue to the dataset, which were not available in this study.

The developed network can be directly applied to a WSI, without pre-processing (i.e., selecting a specific region-of-interest). Since the spatial distribution of tumor budding is important, our approach allows larger areas to be scored, something that was not possible with visual assessment, which might give insights in tumor heterogeneity. To investigate this, we calculated the entropy values on the machine-detected TB in the whole slide images (density maps) and found the highest impact with the patient survival data here with the single (median) cut-off value of 0.77. Future work on the spatial relationship of tumor buds should provide more insight in tumor heterogeneity. We graphically displayed the human and machine-produced TB counts per hotspot in Figure 5B, illustrating a deviation from the expert scoring starting in the presence of more than 20 TB per hotspot. We hypothesize that, because of the increased density, counting becomes more complicated for the pathologist, who might ignore doubtful TB and difficult areas (e.g., near ruptured glands, as shown in Figure 4C,D). The upper limit of 10 TB in the current ITBCC scoring system also might lead to higher discrepancy between the algorithm and a pathologist in case of many TB. Despite the fairly strong correlation (see Section 3.3), the TB counts of the student faster R-CNN network within the hand-selected areas from the dataset are on average 1.3 times higher than the numbers reported by the expert. A previous study [22] showed significant discrepancy between the human observers, with a variation between pathologists of up to 1.8 times—the number of TBs within the same hotspot. The observed variation with the algorithm is thus in line with human interobserver variability. Other algorithms [4] based on CK-positive objects detected approximately 2.5 times more buds than with a manual count.

A good match was found between the network-selected top 3 and the expert-selected hotspots in almost half of the number of the test images, and approximately three of the four hand-selected hotspots matched at least 1 of the ten automatically generated samples. Where no overlap was found, we see cases where the algorithm selects locations in the main tumor mass, where manually selected hotspots are located in the invasive front area in accordance with the ITBCC guideline. Although the prognostic significance of intratumoral TB is uncertain, Pai et al. [23], in a recent study, found almost equal associations with outcome data for WSI segmented proportions TB/PDC (taken as one feature) in the tumor edge region and for the TB/PDC proportions in the main tumor mass, suggesting that there is also a role for intratumoral TB/PDC. Other reasons for these ‘full mismatch’ cases are that these are caused by the poor images quality, for example, artifacts as a result of mechanical processing, which should be avoided with better quality control (see Figure 5C,D). Observer variability in hotspot selection was not considered in this study. Further research should determine whether this variability is reduced when pathologists are presented with a TB density map, as shown in Figure 5A. Using the clinical data of a small group of patients, we investigated the relationship between the outcome and tumor buds on survival. We found that all methods (manual and automatic) show a similar effect on survival based on the measured amount of tumor buds. Large, multicenter validation studies can now commence in order to determine the validation before the implementation of this algorithm.

## 5. Conclusions

We developed a fully automatic TB detection system that obtained high correlation with manual TB assessment. With a view to possible clinical application, it could primarily be used for the selection—and therefore also registration—of the hotspot location, which could already partly solve the problems with reproducibility. The automatic detection of TB at the WSI scale makes it possible to determine and further investigate promising indicators, such as entropy values derived from TB numbers as a biomarker, especially since we made the model publicly available for research use.

## 6. Code Availability

The faster-RCNN student model is available for research use on https://grand-challenge.org/algorithms/colon-budding-in-ihc/, accessed on 1 February 2023).

## Figures and Tables

**Figure 1 cancers-15-02079-f001:**
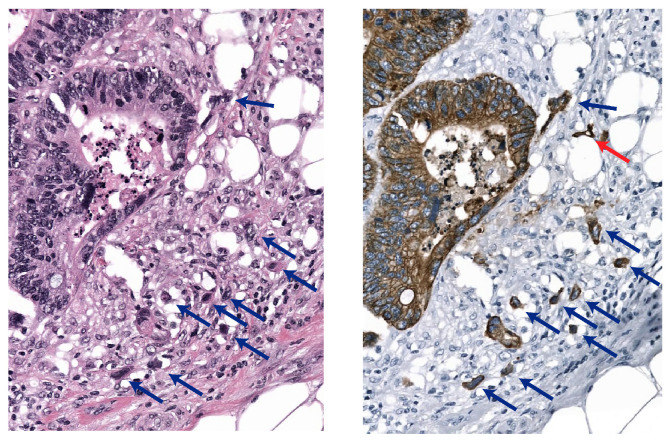
Tumor buds in a corresponding H&E-stained (**left**) and pan-CK stained (**right**) image from the same histology slide after destaining. Note the red arrow that shows a tumor bud that is not visible in the H&E-stained image, buds that are visible in both stains are shown by the blue arrows.

**Figure 2 cancers-15-02079-f002:**
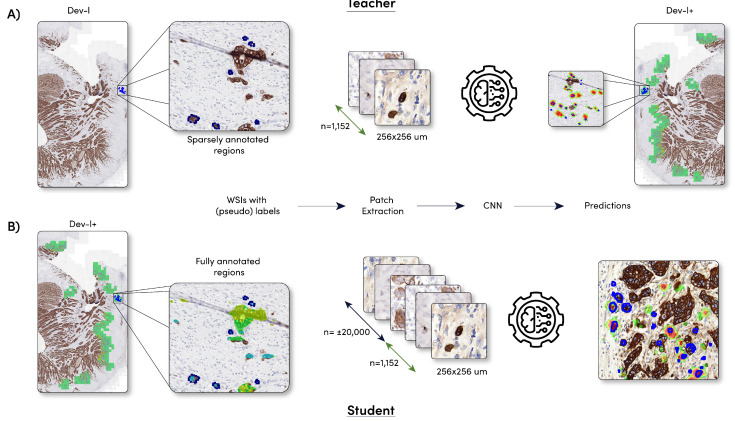
Schematic overview of the model development. (**A**) Training of the two teacher models (DenseNet and faster R-CNN) and the sparse dataset dev-l. Note that the 1152 candidates consist of 680 + 472 tumor bud/non-tumor-bud candidates, respectively; (**B**) training of the student models (DenseNet and faster R-CNN) and the fully exhaustive dataset dev-l+. Dataset dev-l+ was created by applying the teacher model to dataset dev-l.

**Figure 3 cancers-15-02079-f003:**
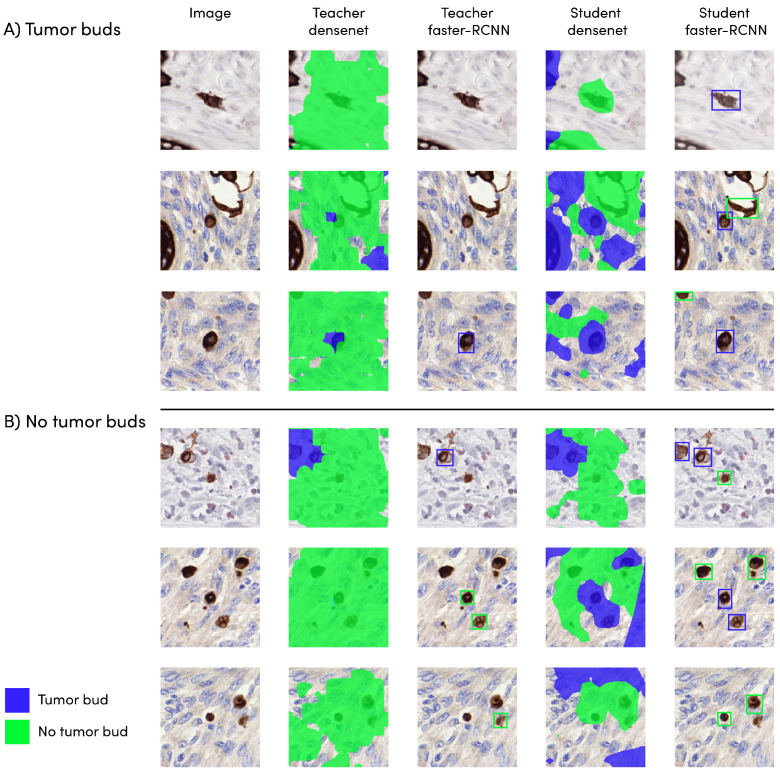
Model output on (**A**) tumor bud and (**B**) no tumor bud. The pixels in green/blue indicate the automatically predicted tumor bud/non-tumor-bud regions. The reference standard labels are related to the object in the center of the image.

**Figure 4 cancers-15-02079-f004:**
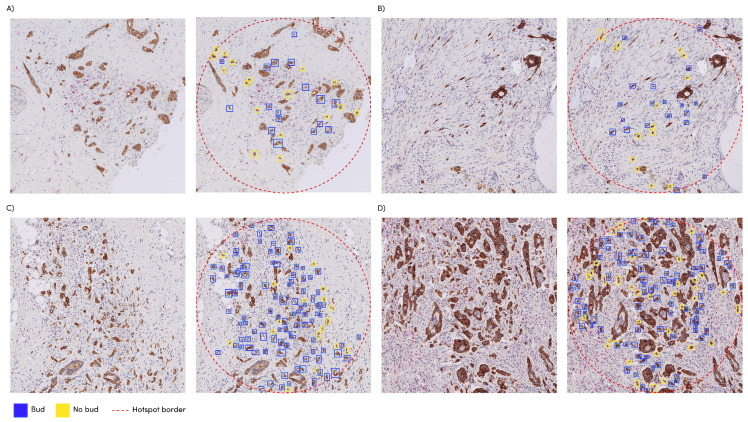
Manual selected hotspots with the automatic detections as overlay: (**A**) manual count: 13, automatic count: 17; (**B**) manual count: 20, automatic count: 15; (**C**) manual count: 23, automatic count: 86; (**D**) manual count: 24, automatic count: 72.

**Figure 5 cancers-15-02079-f005:**
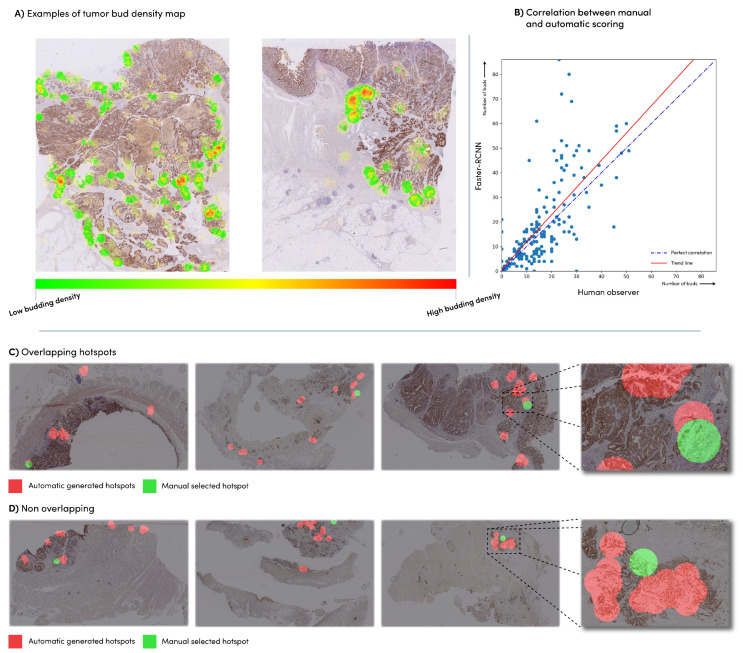
(**A**) Examples of a density heatmap generated based on the automatic tumor bud detections; (**B**) correlation between the manual tumor bud counts in the manual hotspot and the automatic count in the same hotspot; (**C**) Manually selected hotspots (green) vs. automatic hotspots (red). (**A**) cases with overlap; and (**D**) cases with no overlap.

**Table 1 cancers-15-02079-t001:** Overview of the different datasets used in this study.

Dataset	Number of WSIs	Origin (# of Slides)	Annotations
dev-l	51	Bern (3), Dublin (1), Nijmegen (47)	480 tumor bud candidates and 321 non-tumor-bud candidates
dev-v	23	Bern (1), Dublin (0), Nijmegen (21)	200 tumor bud candidates and 151 non-tumor-bud candidates
dev-t	10	Bern (2), Dublin (1), Nijmegen (8)	330 tumor bud candidates and 283 non-tumor-bud candidates
eval	240	Bern (240)	Manual hotspot locations and number of tumor buds within this hotspot

**Table 2 cancers-15-02079-t002:** Clinicopathological characteristics of the patient cohort eval for analysis. Note: because of the limited number of patients within every group, we merged T-stages 1 and 2; 3 and 4; and N 0, 1, and 2.

		n	%
Sex	Male	15	62.5
	Female	9	37.5
Age, years	<65	14	58.3
	≥65	10	41.6
Invasion depth	T1T2	7	29.1
	T3T4	17	70.9
Nodal status	0 and 1	18	75.0
	2	6	25.0
Death	Yes	14	58.3
	No	10	41.6

**Table 3 cancers-15-02079-t003:** Sensitivity scores of different teacher and student models on dataset dev-t. The overall best performing model was highlighted.

	Model	Sensitivity
Teacher	DenseNet	0.83
	Faster R-CNN	0.47
Student	DenseNet	0.87
	Faster R-CNN	0.91

## Data Availability

The data presented in this study are available on request from the corresponding author.

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
