# Peer review of "Semi-Supervised Learning to Automate Tumor Bud Detection in Cytokeratin-Stained Whole-Slide Images of Colorectal Cancer"

_cancers, 2023, doi:10.3390/cancers15072079_

Round 1

Reviewer 1 Report

The submitted manuscript presents an interesting method of semi-supervising learning for automated detection of tumor buds in cytokeratin-stamned whole-slide images of CRC.

The model was developed by training of two teacher models (Densenet & Faster R-CNN) and two student models (Densenet & Faster R-CNN) as shown on the Figure 2. The results of model output on tumor bud and no tumor bud (Figure 3) are much better for for Faster R-CNN  than Densenet in teacher and student model respectively. 

However, I am concerned about significant discrepancies between numbers of buds detected by a human observer and Faster R-CNN method (AI). This should be explained carefully in the presented manuscript (Figures 4 and 5). 

The algorithm 2.2.4. should be described step by step in a formal way. In the manuscript the description of the algorithm is mixed with the Authors remarks. The remarks should be included in a separate section of the paper.  

The description of Table 2 is not consistent in nodal status (0 and 12 in the table, while 0&1 and 2 in the table description).

The Discussion is poor, this is a self discussion rather than a confrontation with other publications on machine learning methods and artificial intelligence in histopathology. Perhaps, it will be better to name this part of the paper as "Final comments" instead of "Discussion" or extend the Discussion. 

Conclusion is missing in this paper. The section "Conclusion" presents the achievement of the Authors rather than a real conclusion. 

Reviewer 2 Report

This manuscript described a unique automatic system detecting tumor budding in colorectal carcinomas. The background of this study was well-written. The results of this study are very interesting, and would provide special interest to journal readers. 

Author Response

Thank you very much for your positive feedback. It’s very much appreciated.

Reviewer 3 Report

I am really grateful for reviewing this manuscript. In my opinion, this manuscript can be published once some revision is done successfully. This study used Faster RCNN and 324 cytokeratin-stained whole-slide images of colorectal cancer to achieve the sensitivity of 91% for the automatic detection of tumor buds. I would argue that this is a rare achievement. However, it needs to be noted that Single Shot Detector is reported to be faster than Faster RCNN with comparable performance for object detection. I would like to ask the authors to address this issue in the section of Discussion. 
